# ADAPTIVE NEURAL TREES

## ABSTRACT

Deep neural networks and decision trees operate on largely separate paradigms; typically, the former performs representation learning with pre-specified architectures, while the latter is characterised by learning hierarchies over pre-specified features with data-driven architectures. We unite the two via adaptive neural trees (ANTs), a model that incorporates representation learning into edges, routing functions and leaf nodes of a decision tree, along with a backpropagation-based training algorithm that adaptively grows the architecture from primitive modules (e.g., convolutional layers). ANTs allow increased interpretability via hierarchical clustering, e.g., learning meaningful class associations, such as separating natural vs. man-made objects. We demonstrate this on classification and regression tasks, achieving over 99% and 90% accuracy on the MNIST and CIFAR-10 datasets, and outperforming standard neural networks, random forests and gradient boosted trees on the SARCOS dataset. Furthermore, ANT optimisation naturally adapts the architecture to the size and complexity of the training data.

## 1 INTRODUCTION

Neural networks (NNs) and decision trees (DTs) are both powerful classes of machine learning models with proven successes in academic and commercial applications. The two approaches, however, typically come with mutually exclusive benefits and limitations.

NNs are characterised by learning hierarchical representations of data through the composition of nonlinear transformations (Zeiler & Fergus, 2014; Bengio, 2013), which has alleviated the need for feature engineering, in contrast with many other machine learning models. In addition, NNs are trained with stochastic optimisers, such as stochastic gradient descent (SGD), allowing training to scale to large datasets. Consequently, with modern hardware, we can train NNs of many layers on large datasets, solving numerous problems ranging from object detection to speech recognition with unprecedented accuracy (LeCun et al., 2015). However, their architectures typically need to be designed by hand and fixed per task or dataset, requiring domain expertise (Zoph & Le, 2017). Inference can also be heavy-weight for large models, as each sample engages every part of the network, i.e., increasing capacity causes a proportional increase in computation (Bengio et al., 2013).

Alternatively, DTs are characterised by learning hierarchical clusters of data (Criminisi & Shotton, 2013). A DT learns how to split the input space, so that in each subset, linear models suffice to explain the data. In contrast to standard NNs, the architectures of DTs are optimised based on training data, and are particularly advantageous in data-scarce scenarios. DTs also enjoy lightweight inference as only a single root-to-leaf path on the tree is used for each input sample. However, successful applications of DTs often require hand-engineered features of data. We can ascribe the limited expressivity of single DTs to the common use of simplistic routing functions, such as splitting on axis-aligned features. The loss function for optimising hard partitioning is non-differentiable, which hinders the use of gradient descent-based optimization and thus complex splitting functions. Current techniques for increasing capacity include ensemble methods such as random forests (RFs) (Breiman, 2001) and gradient-boosted trees (GBTs) (Friedman, 2001), which are known to achieve state-of-the-art performance in various tasks, including medical imaging and financial forecasting (Sandulescu & Chiru, 2016; Kaggle.com, 2017; Le Folgoc et al., 2016; Volkovs et al., 2017).

The goal of this work is to combine NNs and DTs to gain the complementary benefits of both approaches. To this end, we propose *adaptive neural trees* (ANTs), which generalise previous work that attempted the same unification (Suárez & Lutsko, 1999; İrsoy et al., 2012; Laptev & Buhmann,

2014; Rota Bulo & Kontschieder, 2014; Kontschieder et al., 2015; Frosst & Hinton, 2017; Xiao, 2017) and address their limitations (see Tab. 1). ANTs represent routing decisions and root-to-leaf computational paths within the tree structures as NNs, which lets them benefit from hierarchical representation learning, rather than being restricted to partitioning the raw data space. In addition, we propose a backpropagation-based training algorithm to grow ANTs based on a series of decisions between making the ANT deeper—the central NN paradigm—or partitioning the data—the central DT paradigm (see Fig. 1 (Right)). This allows the architectures of ANTs to adapt to the data available. By our design, ANTs inherit the following desirable properties from both DTs and NNs:

- **Representation learning**: as each root-to-leaf path in an ANT is a NN, features can be learnt end-to-end with gradient-based optimisation. This, in turn, allows for learning complex data partitioning. The training algorithm is also amenable to SGD.
- **Architecture learning**: by progressively growing ANTs, the architecture adapts to the availability and complexity of data, embodying Occams razor. The growth procedure can be viewed as architecture search with a hard constraint over the model class.
- **Lightweight inference**: at inference time, ANTs perform conditional computation, selecting a single root-to-leaf path on the tree on a per-sample basis, activating only a subset of the parameters of the model.

We empirically validate these benefits for classification and regression through experiments on the MNIST (LeCun et al., 1998), CIFAR-10 (Krizhevsky & Hinton, 2009) and SARCOS (Vijayakumar & Schaal, 2000) datasets. Along with other forms of neural networks, ANTs far outperform state-of-the-art random forest (RF) (Zhou & Feng, 2017) and gradient boosted tree (GBT) (Ponomareva et al., 2017) methods on the image-based classification datasets, with architectures achieving over 99% accuracy on MNIST and over 90% accuracy on CIFAR-10. On the other hand, the best performing methods on the SARCOS multivariate regression dataset are all tree-based, with soft decision trees (SDTs) (Suárez & Lutsko, 1999; Jordan & Jacobs, 1994), GBTs (Friedman, 2001) and ANTs achieving the lowest mean squared error. At the same time, ANTs can learn meaningful hierarchical partitionings of data, e.g., grouping man-made and natural objects (see Fig. 2). ANTs also have reduced time and memory requirements during inference, conferred by conditional computation. In one case, we discover an architecture that achieves over $98\%$ accuracy on MNIST using approximately the same number of parameters as a linear classifier on raw image pixels, showing the benefits of modelling a hierarchical structure that reflects the underlying data structure in enhancing both computational and predictive performance. Finally, we demonstrate the benefits of architecture learning by training ANTs on subsets of CIFAR-10 of varying sizes. The method can construct architectures of adequate size, leading to better generalisation, particularly on small datasets.

## 2 RELATED WORK

Our work is primarily related to research into combining DTs and NNs to benefit from the power of representation learning. Here we explain how ANTs subsumes a large body of such prior work as specific cases and address their limitations. We include additional reviews of work in conditional computation and neural architecture search in Sec. B in the supplementary material.

The very first SDT introduced in (Suárez & Lutsko, 1999) is a specific case where in our terminology the routers are axis-aligned features, the transformers are identity functions, and the routers are static distributions over classes or linear functions. The hierarchical mixture of experts (HMEs) proposed by (Jordan & Jacobs, 1994) is a variant of SDTs whose routers are linear classifiers and the tree structure is fixed. More modern SDTs in (Rota Bulo & Kontschieder, 2014; Laptev & Buhmann, 2014; Frosst & Hinton, 2017) used multilayer perceptrons (MLPs) or convolutional layers in the routers to learn more complex partitionings of the input space. However, the simplicity of identity transformers used in these methods means that input data is never transformed and thus each path on the tree does not perform representation learning, limiting their performance.

More recent work suggested that integrating non-linear transformations of data into DTs would enhance model performance. The neural decision forest (NDF) (Kontschieder et al., 2015), which held cutting-edge performance on ImageNet (Deng et al., 2009) in 2015, is an ensemble of DTs, each of which is also an instance of ANTs where the whole GoogLeNet architecture (Szegedy et al., 2015) (except for the last linear layer) is used as the root transformer, prior to learning tree-structured

Table 1: Comparison of tree-structured NNs. The first column denotes if each path on the tree is a NN, and the second column denotes if the routers learn features from data. The last column indicates if the method grows an architecture, or uses a pre-specified one.

| Method | Feature learning? | | Grown? |
| --- | --- | --- | --- |
| | Path | Routers | |
| SDT (Suárez & Lutsko, 1999) | ✗ | ✗ | ✓ |
| SDT 2 / HME (Jordan & Jacobs, 1994) | ✗ | ✓ | ✗ |
| SDT 3 (İrsoy et al., 2012) | ✗ | ✓ | ✓ |
| SDT 4 (Frosst & Hinton, 2017) | ✗ | ✓ | ✗ |
| BT (İrsoy et al., 2014) | ✗ | ✓ | ✓ |
| Conv DT (Laptev & Buhmann, 2014) | ✗ | ✓ | ✗ |
| NDT (Rota Bulo & Kontschieder, 2014) | ✗ | ✓ | ✓ |
| NDT 2 (Xiao, 2017) | ✓ | ✓ | ✗ |
| NDF (Kontschieder et al., 2015) | ✓ | ✓ | ✗ |
| CNet (Ioannou et al., 2016) | ✓ | ✓ | ✗ |
| **ANT (ours)** | ✓ | ✓ | ✓ |

classifiers with linear routers. Xiao (2017) employed a similar approach with a MLP at the root transformer, and is optimised to minimise a differentiable information gain loss. The conditional network proposed in (Ioannou et al., 2016) sparsified CNN architectures by distributing computations on hierarchical structures based on directed acyclic graphs with MLP-based routers, and designed models with the same accuracy with reduced compute cost and number of parameters. However, in all cases, the model architectures are pre-specified and fixed.

In contrast, ANTs satisfy all criteria in Tab. 1; they provide a general framework for learning tree-structured models with the capacity of representation learning along each path and within routing functions, and a mechanism for learning its architecture.

Architecture growth is a key facet of DTs (Criminisi & Shotton, 2013), and typically performed in a greedy fashion with a termination criteria based on validation set error (Suárez & Lutsko, 1999; İrsoy et al., 2012). Here we review previous attempts to improve upon this greedy growth strategy in the DT literature. Decision jungles (Shotton et al., 2013) employ a training mechanism to merge partitioned input spaces between different sub-trees, and thus to rectify suboptimal "splits" made due to the locality of optimisation. İrsoy et al. (2014) proposes budding trees, which are grown and pruned incrementally based on global optimisation of all existing nodes. While our proposed training algorithm, for simplicity, grows the architecture by greedily choosing the best option between going deeper and splitting the input space (see Fig. 1), it is certainly amenable to the above advances.

Another related strand of work for feature learning is cascaded forests—stacks of RFs where the outputs of intermediate models are fed into the subsequent ones (Montillo et al., 2011; Kontschieder et al., 2013; Zhou & Feng, 2017). It has been shown how a cascade of DTs can be mapped to NNs with sparse connections (Sethi, 1990), and more recently Richmond et al. (2015) extended this argument to RFs. However, the features obtained in this approach are the intermediate outputs of respective component models, which are not optimised for the target task, and cannot be learned end-to-end, thus limiting its representational quality.

## 3 ADAPTIVE NEURAL TREES

We now formalise the definition of Adaptive Neural Trees (ANTs), which are a form of DTs enhanced with deep, learned representations. We focus on supervised learning, where the aim is to learn the conditional distribution $p(\mathbf{y}|\mathbf{x})$ from a set of $N$ labelled samples $(\mathbf{x}^{(1)}, \mathbf{y}^{(1)}), ..., (\mathbf{x}^{(N)}, \mathbf{y}^{(N)}) \in \mathcal{X} \times \mathcal{Y}$ as training data.

### 3.1 MODEL TOPOLOGY AND OPERATIONS

In short, an ANT is a tree-structured model, characterized by a set of hierarchical partitions of the input space $\mathcal{X}$, a series of nonlinear transformations, and separate predictive models in the respective component regions. More formally, we define an ANT as a pair $(\mathbb{T}, \mathbb{O})$ where $\mathbb{T}$ defines the model topology, and $\mathbb{O}$ denotes the set of operations on it.

We restrict the model topology $\mathbb{T}$ to be instances of *binary trees*, defined as a set of finite graphs where every node is either an internal node or a leaf, and is the child of exactly one parent node

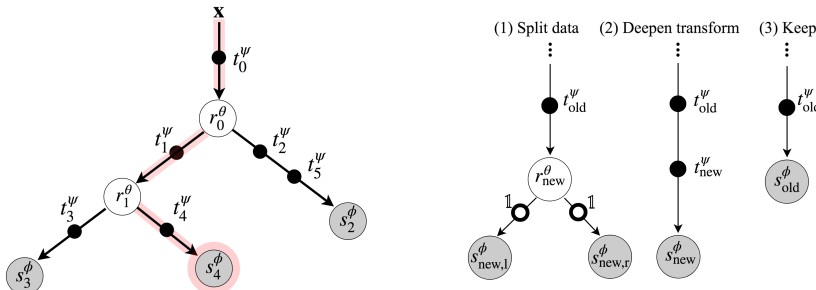

Figure 1: **(Left).** An example of an ANT architecture. Data is passed through transformers (black circles on edges), routers (white circles on internal nodes), and solvers (gray circles on leaf nodes). The red shaded path shows routing of $\mathbf{x}$ to reach leaf node 4. Input $\mathbf{x}$ undergoes a series of selected transformations $\mathbf{x} \to \mathbf{x}_0^{\psi} := t_0^{\psi}(\mathbf{x}) \to \mathbf{x}_1^{\psi} := t_1^{\psi}(\mathbf{x}_0^{\psi}) \to \mathbf{x}_4^{\psi} := t_4^{\psi}(\mathbf{x}_1^{\psi})$ and the solver module yields the predictive distribution $p_4^{\phi,\psi}(\mathbf{y}) := s_4^{\phi}(\mathbf{x}_4^{\psi})$. The probability of selecting this path is given by $\pi_2^{\psi,\theta}(\mathbf{x}) := r_0^{\theta}(\mathbf{x}_0^{\psi}) \cdot (1 - r_1^{\theta}(\mathbf{x}_1^{\psi}))$. **(Right).** Three growth options at a given node: *split data, deepen transform & keep*. The small white circles on the edges denote identity transformers.

(apart from the parent-less root node). We define the topology of a tree as $\mathbb{T} := \{\mathcal{N}, \mathcal{E}\}$ where $\mathcal{N}$ is the set of all nodes, and $\mathcal{E}$ is the set of edges between them. Nodes with no children are leaf nodes, $\mathcal{N}_{leaf}$, and all others are internal nodes, $\mathcal{N}_{int}$. Every internal node $j \in \mathcal{N}_{int}$ has exactly two children nodes, represented by $\text{left}(j)$ and $\text{right}(j)$. Unlike standard trees, $\mathcal{E}$ contains an edge which connects input data $\mathbf{x}$ with the root node, as shown in Fig.1 (Left).

Every node and edge is assigned with operations which acts on the allocated samples of data (Fig.1). Starting at the root, each sample gets transformed and traverses the tree according to the set of operations $\mathbb{O}$. An ANT is constructed based on three primitive modules of differentiable operations:

1. **Routers**, $\mathcal{R}$: each internal node $j \in \mathcal{N}_{int}$ holds a *router* module, $r_j^{\theta} : \mathcal{X}_j \to [0, 1] \in \mathcal{R}$, parametrised by $\theta$, which sends samples from the incoming edge to either the left or right child. Here $\mathcal{X}_j$ denotes the representation at node $j$. We use *stochastic routing*, where the binary decision (1 for the left and 0 for the right branch) is sampled from Bernoulli distribution with mean $r_j^{\theta}(\mathbf{x}_j)$ for input $\mathbf{x}_j \in \mathcal{X}_j$. As an example, $r_j^{\theta}$ can be defined as a small convolutional neural network (CNN).

2. **Transformers**, $\mathcal{T}$: every edge $e \in \mathcal{E}$ of the tree has one or a composition of multiple *transformer* module(s). Each transformer $t_e^{\psi} \in \mathcal{T}$ is a nonlinear function, parametrised by $\psi$, that transforms samples from the previous module and passes them to the next one. For example, $t_e^{\psi}$ can be a single convolutional layer followed by ReLU (Nair & Hinton, 2010). Unlike in standard DTs, edges transform data and are allowed to "grow" by adding more operations (Sec. 4), learning "deeper" representations as needed.

3. **Solvers**, $\mathcal{S}$: each leaf node $l \in \mathcal{N}_{leaf}$ is assigned to a *solver* module, $s_l^{\phi} : \mathcal{X}_l \to \mathcal{Y} \in \mathcal{S}$, parametrised by $\phi$, which operates on the transformed input data and outputs an estimate for the conditional distribution $p(\mathbf{y}|\mathbf{x})$. For classification tasks, we can define, for example, $s^{\phi}$ as a linear classifier on the feature space $\mathcal{X}_l$, which outputs a distribution over classes.

Defining operations on the graph $\mathbb{T}$ amounts to a specification of the triplet $\mathbb{O} = (\mathcal{R}, \mathcal{T}, \mathcal{S})$. For example, given image inputs, we would choose the operations of each module to be from the set of operations commonly used in CNNs (examples are given in Tab. 2). In this case, every computational path on the resultant ANT, as well as the set of routers that guide inputs to one of these paths, are given by CNNs. In Sec. 4, we discuss methods for constructing such tree-shaped NNs end-to-end from simple building blocks. Lastly, many existing tree-structured models (Suárez & Lutsko, 1999; İrsoy et al., 2012; Laptev & Buhmann, 2014; Rota Bulo & Kontschieder, 2014; Kontschieder et al., 2015; Frosst & Hinton, 2017; Xiao, 2017) are instantiations of ANTs with limitations which we will address with our model (see Sec. 2 for a more detailed discussion).

## 3.2 PROBABILISTIC MODEL AND INFERENCE

An ANT $(\mathbb{T}, \mathbb{O})$ models the conditional distribution $p(\mathbf{y}|\mathbf{x})$ as a HME (Jordan & Jacobs, 1994), each of which is defined as a NN and corresponds to a particular root-to-leaf path in the tree. The key

Table 2: Primitive module specification for ANTs. The $1^{\text{st}}$ & $2^{\text{nd}}$ rows describe modules for MNIST and CIFAR-10. "conv5-40" denotes a 2D convolution with 40 kernels of spatial size $5 \times 5$. "GAP", "FC" and "LC" stand for global-average-pooling, fully connected layer and linear classifier, respectively. "Downsample Freq" denotes the frequency at which $2 \times 2$ max-pooling is applied.

| Model | Router, $\mathcal{R}$ | Transformer, $\mathcal{T}$ | Solver, $\mathcal{S}$ | Downsample Freq. |
|---|---|---|---|---|
| ANT-MNIST-A | $1 \times$ conv5-40 + GAP + 2×FC | $1 \times$ conv5-40 | LC | 1 |
| ANT-MNIST-B | $1 \times$ conv3-40 + GAP + 2×FC | $1 \times$ conv3-40 | LC | 2 |
| ANT-MNIST-C | $1 \times$ conv5-5 + GAP + 2×FC | $1 \times$ conv5-5 | LC | 2 |
| ANT-CIFAR10-A | $2 \times$ conv3-128 + GAP + 1×FC | $2 \times$ conv3-128 | LC | 1 |
| ANT-CIFAR10-B | $2 \times$ conv3-96 + GAP + 1×FC | $2 \times$ conv3-96 | LC | 1 |
| ANT-CIFAR10-C | $2 \times$ conv3-72 + GAP + 1×FC | $2 \times$ conv3-72 | GAP + LC | 1 |

difference with traditional HMEs is that the input is not only routed but also transformed within the tree hierarchy. Each input $\mathbf{x}$ stochastically traverses the tree based on decisions of routers and undergoes a sequence of selected transformations until it reaches a leaf node where the corresponding solver module predicts the label $\mathbf{y}$. Supposing we have $L$ leaf nodes, the full predictive distribution is given by

$$p(\mathbf{y}|\mathbf{x}, \boldsymbol{\psi}, \boldsymbol{\phi}, \boldsymbol{\theta}) = \sum_{\mathbf{z}} p(\mathbf{y}, \mathbf{z}|\mathbf{x}, \boldsymbol{\theta}, \boldsymbol{\psi}, \boldsymbol{\phi}) = \sum_{l=1}^{L} \underbrace{p(z_l = 1|\mathbf{x}, \boldsymbol{\theta}, \boldsymbol{\psi})}_{\text{Leaf-assignment prob. } \pi_l^{\boldsymbol{\theta}, \boldsymbol{\psi}}} \cdot \underbrace{p(\mathbf{y}|\mathbf{x}, z_l = 1, \boldsymbol{\phi}, \boldsymbol{\psi})}_{\text{Leaf-specific prediction. } p_l^{\boldsymbol{\phi}, \boldsymbol{\psi}}},$$

where $\mathbf{z} \in \{0,1\}^L$ is an $L$-dimensional binary latent variable such that $\sum_{l=1}^{L} z_l = 1$, which describes the choice of leaf node (e.g. $z_l = 1$ means that leaf $l$ is used). Here $\boldsymbol{\theta}, \boldsymbol{\psi}, \boldsymbol{\phi}$ summarise the parameters of router, transformer and solver modules in the tree. The mixing coefficient $\pi_l^{\boldsymbol{\theta}, \boldsymbol{\psi}}(\mathbf{x}) := p(z_l = 1|\mathbf{x}, \boldsymbol{\psi}, \boldsymbol{\theta})$ quantifies the probability that $\mathbf{x}$ is assigned to leaf $l$ and is given by a product of decision probabilities over all router modules on the unique path $\mathcal{P}_l$ from the root to leaf node $l$:

$$\pi_l^{\boldsymbol{\psi}, \boldsymbol{\theta}}(\mathbf{x}) = \prod_{r_j^{\boldsymbol{\theta}} \in \mathcal{P}_l} r_j^{\boldsymbol{\theta}}(\mathbf{x}_j^{\boldsymbol{\psi}})^{\mathbb{1}_{l \swarrow j}} \cdot \left(1 - r_j^{\boldsymbol{\theta}}(\mathbf{x}_j^{\boldsymbol{\psi}})\right)^{1 - \mathbb{1}_{l \swarrow j}}, \tag{1}$$

where $l \swarrow j$ is a binary relation and is only true if leaf $l$ is in the left subtree of internal node $j$, and $\mathbf{x}_j^{\boldsymbol{\psi}}$ is the feature representation of $\mathbf{x}$ at node $j$. Let $\mathcal{T}_j = \{t_{e_1}^{\boldsymbol{\psi}}, ..., t_{e_n}^{\boldsymbol{\psi}}\}$ denote the ordered set of the $n$ transformer modules on the path from the root to node $j$, then the feature vector $\mathbf{x}_j^{\boldsymbol{\psi}}$ is given by

$$\mathbf{x}_j^{\boldsymbol{\psi}} := \left(t_{e_n}^{\boldsymbol{\psi}} \circ ... \circ t_{e_2}^{\boldsymbol{\psi}} \circ t_{e_1}^{\boldsymbol{\psi}}\right)(\mathbf{x}).$$

On the other hand, the leaf-specific conditional distribution $p_l^{\boldsymbol{\phi}, \boldsymbol{\psi}}(\mathbf{y}) := p(\mathbf{y}|\mathbf{x}, z_l = 1, \boldsymbol{\phi}, \boldsymbol{\psi})$ in eq. equation 1 yields an estimate for the distribution over target $\mathbf{y}$ for leaf node $l$ and is given by its solver's output $s_l^{\boldsymbol{\phi}}(\mathbf{x}_{\text{parent}(l)}^{\boldsymbol{\psi}})$.

We consider two schemes of inference, based on a trade-off between accuracy and computation. Firstly, the *full predictive distribution* given in eq. equation 1 is used as the estimate for the target conditional distribution $p(\mathbf{y}|\mathbf{x})$. However, averaging the distributions over all the leaves, weighted by their respective path probabilities, involves computing all operations at all nodes and edges of the tree, which makes inference expensive for a large ANT. We therefore consider a second scheme which uses the predictive distribution at the leaf node chosen by greedily traversing the tree in the directions of highest confidence of the routers. This approximation constrains computations to a single path, allowing for more memory- and time-efficient inference.

## 4 OPTIMISATION

Training of an ANT proceeds in two stages: 1) *growth phase* during which the model architecture is learned based on *local* optimisation, and 2) *refinement phase* which further tunes the parameters of the model discovered in the first phase based on *global* optimisation. We include pseudocode for the joint training algorithm in Sec. A in the supplementary material.

### 4.1 Loss function: optimising parameters for fixed architecture

For both phases, we use the negative log-likelihood (NLL) as the common objective function to minimise, which is given by $-\log p(\mathbf{Y}|\mathbf{X}, \boldsymbol{\theta}, \boldsymbol{\psi}, \boldsymbol{\phi}) = -\sum_{n=1}^{N} \log \left( \sum_{l=1}^{L} \pi_l^{\boldsymbol{\theta}, \boldsymbol{\psi}}(\mathbf{x}^{(n)}) \, p_l^{\boldsymbol{\phi}, \boldsymbol{\psi}}(\mathbf{y}^{(n)}) \right)$ where $\mathbf{X} = \{\mathbf{x}^{(1)}, ..., \mathbf{x}^{(N)}\}$, $\mathbf{Y} = \{\mathbf{y}^{(1)}, ..., \mathbf{y}^{(N)}\}$ denote the training inputs and targets. As all component modules (routers, transformers and solvers) are differentiable with respect to their parameters $\Theta = (\boldsymbol{\theta}, \boldsymbol{\psi}, \boldsymbol{\phi})$, we can use gradient-based optimisation. Given an ANT with fixed topology $\mathbb{T}$, we use backpropagation (Rumelhart et al., 1986) for gradient computation and use gradient descent to minimise the NLL for learning the parameters.

### 4.2 Growth phase: learning architecture $\mathbb{T}$

We next describe our proposed method for growing the tree $\mathbb{T}$ to an architecture of adequate complexity for the availability of training data. Starting from the root, we choose one of the leaf nodes in breadth-first order and incrementally modify the architecture by adding extra computational modules to it. In particular, we evaluate 3 choices (Fig. 1 (Right)) at each leaf node; (1)."split data" extends the current model by splitting the node with an addition of a new router; (2) "deepen transform" increases the depth of the incoming edge by adding a new transformer; (3) "keep" retains the current model. We then locally optimise the parameters of the newly added modules in the architectures of (1) and (2) by minimising NLL via gradient descent, while fixing the parameters of the previous part of the computational graph. Lastly, we select the model with the lowest validation NLL if it improves on the previously observed lowest NLL, otherwise we execute (3) and keep the original model. This process is repeated to all new nodes level-by-level until no more "split data" or "deepen transform" operations pass the validation test.

The rationale for evaluating the two choices is to the give the model a freedom to choose the most effective option between "going deeper" or splitting the data space. Splitting a node is equivalent to a soft partitioning of the feature space of incoming data, and gives birth to two new leaf nodes (left and right children solvers). In this case, the added transformer modules on the two branches are identity functions. Deepening an edge on the other hand does not change the number of leaf nodes, but instead seeks to learn richer representation via an extra nonlinear transformation, and replaces the old solver with a new one.

Local optimisation saves time, memory and compute. Gradients only need to be computed for the parameters of the new peripheral parts of the architecture, reducing the amount of time and computation needed. Forward activations prior to the new parts do not need to be stored in memory, saving space.

### 4.3 Refinement phase: global tuning of $\mathbb{O}$

Once the model topology is determined in the growth phase, we finish by performing global optimisation to refine the parameters of the model, now with a fixed architecture. This time, we perform gradient descent on the NLL with respect to the parameters of all modules in the graph, jointly optimising the hierarchical grouping of data to paths on the tree and the associated expert NNs. The refinement phase can correct suboptimal decisions made during the local optimisation of the growth phase, and empirically improves the generalisation error (see Sec. 5.3).

## 5 Experiments

We evaluate ANTs using the MNIST (LeCun et al., 1998) and CIFAR-10 (Krizhevsky & Hinton, 2009) object classification datasets, and the SARCOS multivariate regression dataset (Vijayakumar & Schaal, 2000) (see Supp. Sec. H for regression and Supp. Sec. I for ensembling details). Here, we first show that ANTs learn hierarchical structures in the data, while still achieving favourable classification accuracies against relevant DT and NN models. Next, we examine the effects of refinement phase on ANTs, and show that it can automatically prune the tree. Finally, we demonstrate that our proposed training procedure adapts the model size appropriately under varying amounts of labelled data. All of our models are constructed using the PyTorch framework (Paszke et al., 2017).

Table 3: Comparison of performance of different models on MNIST and CIFAR-10. The columns "Error (Full)" and "Error (Path)" indicate the classification error of predictions based on the full distribution and the single-path inference. The columns "Params. (Full)" and "Params. (Path)" respectively show the total number of parameters in the model and the average number of parameters utilised during single-path inference. "Ensemble Size" indicates the size of ensemble used to attain the reported accuracy. An entry of "–" indicates that no value was reported. Methods marked with [†] are from our implementations trained in the same experimental setup. * indicates that the parameters are initialised with a pre-trained CNN.

|  | Method | Error % (Full) | Error % (Path) | Params. (Full) | Params. (Path) | Ensemble Size |
|---|---|---|---|---|---|---|
| MNIST | Linear classifier | 7.91 | N/A | **7,840** | N/A | 1 |
|  | Random Forests (Breiman, 2001) | 3.21 | 3.21 | – | – | 200 |
|  | Compact Multi-Class Boosted Trees (Ponomareva et al., 2017) | 2.88 | – | – | – | 100 |
|  | Alternating Decision Forest (Schulter et al., 2013) | 2.71 | 2.71 | – | – | 20 |
|  | Neural Decision Tree (Xiao, 2017) | 2.10 | – | 1,773,130 | 502,170 | 1 |
|  | ANT-MNIST-C | 1.62 | 1.68 | 39,670 | **7,956** | 1 |
|  | MLP with 2 hidden layers (Simard et al., 2003) | 1.40 | N/A | 1,275,200 | N/A | 1 |
|  | LeNet-5[†] (LeCun et al., 1998) | 0.82 | N/A | 431,000 | N/A | 1 |
|  | gcForest (Zhou & Feng, 2017) | 0.74 | 0.74 | – | – | 500 |
|  | ANT-MNIST-B | 0.72 | 0.73 | 76,703 | 50,653 | 1 |
|  | Neural Decision Forest (Kontschieder et al., 2015) | 0.70 | – | 544,600 | 463,180 | 10 |
|  | ANT-MNIST-A | 0.64 | **0.69** | 100,596 | 84,935 | 1 |
|  | CapsNet (Sabour et al., 2017) | **0.25** | – | 8.2M | N/A | 1 |
| CIFAR-10 | Compact Multi-Class Boosted Trees (Ponomareva et al., 2017) | 52.31 | – | – | – | 100 |
|  | Random Forests (Breiman, 2001) | 50.17 | 50.17 | – | – | 2000 |
|  | gcForest (Zhou & Feng, 2017) | 38.22 | 38.22 | – | – | 500 |
|  | MaxOut (Goodfellow et al., 2013) | 9.38 | N/A | 6M | N/A | 1 |
|  | ANT-CIFAR10-C | 9.31 | 9.34 | **0.7M** | **0.5M** | 1 |
|  | ANT-CIFAR10-B | 9.15 | 9.18 | 0.9M | 0.6M | 1 |
|  | Network in Network (Lin et al., 2014) | 8.81 | N/A | 1M | N/A | 1 |
|  | All-CNN[†] (Springenberg et al., 2015) | 8.71 | N/A | 1.4M | N/A | 1 |
|  | ANT-CIFAR10-A | 8.31 | 8.32 | 1.4M | 1.0M | 1 |
|  | ANT-CIFAR10-A* | 6.72 | **6.74** | 1.3M | 0.8M | 1 |
|  | ResNet-110 (He et al., 2016) | 6.43 | N/A | 1.7M | N/A | 1 |
|  | DenseNet-BC (k=40) (Huang et al., 2017) | **3.46** | N/A | 25.6M | N/A | 1 |

## 5.1 PERFORMANCE ON IMAGE CLASSIFICATION

We train ANTs with a range of primitive modules (Tab. 2) and compare against relevant DT and NN models (Tab. 3). In general, DT methods without feature learning, such as RFs (Breiman, 2001; Zhou & Feng, 2017) and GBTs (Ponomareva et al., 2017), perform poorly on complex image data (Krizhevsky & Hinton, 2009). In comparison with CNNs without shortcut connections (LeCun et al., 1998; Goodfellow et al., 2013; Lin et al., 2014; Springenberg et al., 2015), different ANTs balance between strong performance with comparable numbers of trainable parameters, and reasonable performance with a relatively small amount of parameters. At the other end of the spectrum, state-of-the-art NNs (Sabour et al., 2017; Huang et al., 2017) contain significantly more parameters.

For simplicity, we define primitive modules based on three types of NN layers: convolutional, global-average-pooling (GAP) and fully-connected (FC). Solver modules are fixed as linear classifiers (LC) with a softmax output. Router modules are binary classifiers with a sigmoid output. All convolutional and FC layer are followed by ReLUs, except in the last layers of solvers and routers. We also apply $2 \times 2$ max-pooling to feature maps after every $d$ transformer modules where $d$ is the downsample frequency. We balance the number of parameters in the router and transformer modules to be of the same order of magnitude to avoid favouring either partitioning the data or learning more expressive features. We hold out $10\%$ of training images as a validation set, on which the best performing model is selected. Full training details, including training times, are provided in the supplementary material.

**Two inference schemes:** for each ANT, classification is performed in two ways: multi-path inference with the full predictive distribution (eq. equation 1), and single-path inference based on the greedily-selected leaf node (Sec. 3.2). We observed that with our training scheme the splitting probabilities in the routers tend to be very confident, being close to 0 or 1 (see histograms in blue in Fig. 2(b)). This means that single path inference gives a good approximation of the multi-path inference but is more efficient to compute. We show this holds empirically in Tab. 3, where the largest difference between Error (Full) and Error (Path) is $0.06\%$ while number of parameters is reduced from Params (Full) to Params (Path) across all ANT models.

**Patience-based local optimisation:** in the growth phase the parameters for the new modules are trained until convergence, as determined by patience-based early stopping on the validation set. We observe that very low or high patience levels result in new modules underfitting or overfitting locally, respectively, thus preventing meaningful further growth. We tuned this hyperparameter using the validation sets, and set the patience level to 5, which produced consistently good performance on both MNIST and CIFAR-10 datasets across different specifications of primitive modules. A quantitative evaluation is given in the supplementary (Sec. E).

**MNIST digit classification:** we observe that ANT-MNIST-A outperforms state-of-the-art GBT (Ponomareva et al., 2017) and RF (Zhou & Feng, 2017) methods in accuracy. This performance is attained despite the use of a single tree, while RF methods operate with ensembles of classifiers (the size shown in Tab. 2). In particular, the NDF (Kontschieder et al., 2015) has a pre-specified architecture where LeNet-5 (LeCun et al., 1998) is used as the root transformer module, and 10 trees of fixed depth 5 are constructed from this base feature extractor. On the other hand, ANT-MNIST-A is constructed in a data-driven manner from primitive modules, and displays an improvement over the NDF both in terms of accuracy and number of parameters. In addition, reducing the size of convolution kernels (ANT-MNIST-B) reduces the total number of parameters by $25\%$ and the path-wise average by almost $40\%$ while only increasing absolute error by $< 0.1\%$.

We also compare against the LeNet-5 CNN (LeCun et al., 1998), comprised of the same types of operations used in our primitive modules (i.e. convolutional, max-pooling and FC layers). For a fair comparison, the network is trained with the same protocol as that of the ANT refinement phase, achieving an error rate of $0.82\%$ (lower than the reported value of $0.87\%$) on the test set. Both ANT-MNIST-A and ANT-MNIST-B attain better accuracy with a smaller number of parameters than LeNet-5. The current state-of-the-art, capsule networks (CapsNets) (Sabour et al., 2017), have more parameters than ANT-MNIST-A by almost two orders of magnitude.[1] By ensembling ANTs we can reach similar performance ($0.29\%$ versus $0.25\%$; see Tab. 9) with an order of magnitude less parameters (see Tab. 10).

Lastly, we highlight the observation that ANT-MNIST-C, with the simplest primitive modules, achieves an error rate of $1.68\%$ with single-path inference, which is significantly better than that of the linear classifier ($7.91\%$), while engaging almost the same number of parameters ($7,956$ vs. $7,840$) on average. To isolate the benefit of convolutions, we took one of the root-to-path CNNs on ANT-MNIST-C and increased the number of kernels to adjust the number of parameters to the same value. We observe a higher error rate of $3.55\%$, which indicates that while convolutions are beneficial, data partitioning has additional benefits in improving accuracy. This result demonstrates the potential of ANT growth protocol for constructing performant models with lightweight inference. See Sec. G in the supplementary materials for the architecture of ANT-MNIST-C.

**CIFAR-10 object recognition:** we see that variants of ANTs outperform the state-of-the-art DT method, gcForest (Zhou & Feng, 2017) by a large margin, achieving over 90% accuracy, demonstrating the benefit of representation learning in tree-structured models. Secondly, with fewer number of parameters in single-path inference, ANT-CIFAR-A achieves higher accuracy than CNN models without shortcut connections (Goodfellow et al., 2013; Lin et al., 2014; Springenberg et al., 2015) that held the state-of-the-art performance at the time of publication. With simpler primitive modules we learn more compact models (ANT-MNIST-B and -C) with a marginal compromise in accuracy. In addition, initialising the parameters of transformers and routers from a pre-trained single-path CNN further reduced the error rate of ANT-MNIST-A by 20% (see ANT-MNIST-A* in Tab. 3), which indicates room for improvement in our proposed optimisation method.

Shortcut connections (Fahlman & Lebiere, 1990) have recently lead to leaps in performance in deep CNNs (He et al., 2016; Huang et al., 2017). We observe that our best network, ANT-MNIST-A*, has a comparable error rate and half the parameter count (with single-path inference) to the best-performing residual network, ResNet-110 (He et al., 2016). Densely connected networks leads to substantially better accuracy, but with an order of magnitude more parameters (Huang et al., 2017). We expect that shortcut connections could also improve ANT performance, and leave integrating them to future work.

---

[1]Notably, CapsNets also feature a routing mechanism, but with a significantly different mechanism and motivation.

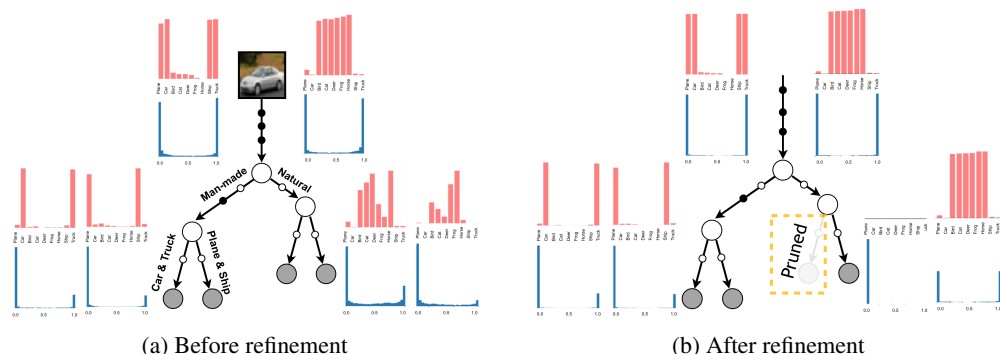

(a) Before refinement         (b) After refinement

Figure 2: Visualisation of class distributions (red) and path probabilities (blue) at respective nodes of an example ANT (a) before and (b) after the refinement phase. (a) shows that the learned model captures an interpretable hierarchy, grouping semantically similar images on the same branches. (b) shows that the refinement phase polarises path probabilities, pruning a branch.

**Ablation study:** we lastly compare the classification errors of different variants of ANTs in cases where the options for adding transformer or router modules are disabled (see Tab. 4). In this experiment, patience levels are tuned separately for respective models. In the first case, the resulting models are equivalent to SDTs (Suárez & Lutsko, 1999) or HMEs (Jordan & Jacobs, 1994) with locally grown architectures, while the second case is equivalent to standard CNNs, grown adaptively layer by layer. We observe that either ablation consistently leads to higher classification errors across different module configurations.

Table 4: Ablation study to compare the effects of different components of ANTs on classification performance. "CNN" refers to the case where the ANT is grown without routers while "SDT/HME" refers to the case where transformer modules on the edges are disabled.

| Module Spec. | Error % (Full) | | | Error % (Path) | | |
|---|---|---|---|---|---|---|
| | ANT (default) | CNN (no routers) | SDT/HME (no transformers) | ANT (default) | CNN (no routers) | SDT/HME (no transformers) |
| ANT-MNIST-A | 0.64 | 0.74 | 3.18 | 0.69 | 0.74 | 4.19 |
| ANT-MNIST-B | 0.72 | 0.80 | 4.63 | 0.73 | 0.80 | 3.62 |
| ANT-MNIST-C | 1.62 | 3.71 | 5.70 | 1.68 | 3.71 | 6.96 |
| ANT-CIFAR10-A | 8.31 | 9.29 | 39.29 | 8.32 | 9.29 | 40.33 |
| ANT-CIFAR10-B | 9.15 | 11.08 | 43.09 | 9.18 | 11.08 | 44.25 |
| ANT-CIFAR10-C | 9.31 | 11.61 | 48.59 | 9.34 | 11.61 | 50.02 |

## 5.2 INTERPRETABILITY

The growth procedure of ANTs is capable of discovering hierarchical structures in the data that are useful to the end task. Learned hierarchies often display strong specialisation of paths to certain classes or categories of data on both the MNIST and CIFAR-10 datasets. Fig. 2 (a) displays an example with particularly "human-interpretable" partitions e.g. man-made versus natural objects, and road vehicles versus other types of vehicles. It should, however, be noted that human intuitions on relevant hierarchical structures do not necessarily equate to optimal representations, particularly as datasets may not necessarily have an underlying hierarchical structure, e.g., MNIST. Rather, what needs to be highlighted is the ability of ANTs to learn when to share or separate the representation of data to optimise end-task performance, which gives rise to automatically discovering such hierarchies. To further attest that the model learns a meaningful routing strategy, we also present the test accuracy of the predictions from the leaf node with the smallest reaching probability in Supp. Sec. F. We observe that using the least likely "expert" leads to a substantial drop in classification accuracy. In addition, we observe that most learned trees are unbalanced (see Supp. Sec. G for more examples). This property of adaptive computation is plausible since certain types of images may be easier to classify than others, as seen in prior work (Figurnov et al., 2017).

## 5.3 EFFECT OF GLOBAL REFINEMENT

We observe that global refinement phase improves the generalisation error. Fig. 3 (Right) shows the generalisation error of various ANT models on CIFAR-10, with vertical dotted lines indicating the epoch when the models enter the refinement phase. As we switch from optimising parts of the ANT in isolation to optimising all parameters, we shift the optimisation landscape, resulting

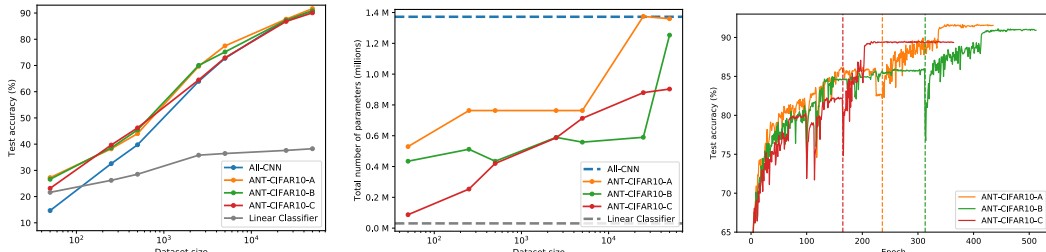

Figure 3: Using CIFAR-10, in (Left), we assess the performance of ANTs for varying amounts of training data. (Middle) The complexity of the grown ANTs increases with dataset size. (Right) Global refinement improves the generalisation; the dotted lines show the epochs at which the models enter the refinement phase.

in an initial drop in performance. However, they all consistently converge to higher test accuracy than the best value attained during the growth phase. This provides evidence that refinement phase remedies suboptimal decisions made during the locally-optimised growth phase. In many cases, we observed that global optimisation polarises the decision probability of routers, which occasionally leads to the effective "pruning" of some branches. For example, in the case of the tree shown in Fig. 2(b), we observe that the decision probability of routers are more concentrated near 0 or 1 after global refinement, and as a result, the empirical probability of visiting one of the leaf nodes, calculated over the validation set, reduces to 0.09%—meaning that the corresponding branch could be pruned without a negligible change in the network's accuracy. The resultant model attains lower generalisation error, showing that the pruning has resolved a suboptimal partioning of data. We emphasise that this is a consequence of global fine-tuning, and does not involve additional algorithms that would be used to prune or compress standard NNs.

## 5.4 ADAPTIVE MODEL COMPLEXITY

Overparametrised models, trained without regularization, are vulnerable to overfitting on small datasets. Here we assess the ability of our proposed ANT training method to adapt the model complexity to varying amounts of labelled data. We run classfication experiments on CIFAR-10 and train three variants of ANTs, the baseline All-CNN (Springenberg et al., 2015) and linear classifier on subsets of the dataset of sizes 50, 250, 500, 2.5k, 5k, 25k and 45k (the full training set). We choose All-CNN as the baseline as it reports the lowest error among the comparison targets and is the closest in terms of constituent operations (convolutional, GAP and FC layers).

Fig.3 (Left) shows the corresponding test performances. The best model is picked based on the performance on the same validation set of 5k examples as before. As the dataset gets smaller, the margin between the test accuracy of the ANT models and All-CNN/linear classifier increases (up to 13%). Fig. 3 (Middle) shows the model size of discovered ANTs as the dataset size varies. It can be observed that for different settings of primitive modules, the number of parameters generally increases as a function of the dataset size. All-CNN has a fixed number of parameters, consistently larger than the discovered ANTs, and suffers from overfitting, particularly on small datasets. The linear classifier, on the other hand, underfits to the data. Our method constructs models of adequate complexity, leading to better generalisation. This shows the added value of our tree-building algorithm over using models of fixed-size structures.

## 6 CONCLUSION

We introduced Adaptive Neural Trees (ANTs), a holistic way to marry the architecture learning, conditional computation and hierarchical clustering of decision trees (DTs) with the hierarchical representation learning and gradient descent optimization of deep neural networks (DNNs). Our proposed training algorithm optimises both the parameters and architectures of ANTs through progressive growth, tuning them to the size and complexity of the training dataset. Together, these properties make ANTs a generalisation of previous work attempting to unite NNs and DTs. Finally, we validated the claimed benefits of ANTs on standard regression and object classification datasets, whilst still achieving high performance.

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

# Adaptive Neural Trees: Supplementary Material

## A  TRAINING ALGORITHM

---

**Algorithm 1** ANT Optimisation

---

Initialise topology $\mathbb{T}$ and parameters $\mathbb{O}$          $\triangleright$ $\mathbb{T}$ is set to a root node with one solver and one transformer
Optimise parameters in $\mathbb{O}$ via gradient descent on NLL          $\triangleright$ Learning root classifier
Set the root node "suboptimal"
**while** true **do**          $\triangleright$ Growth of $\mathbb{T}$ begins
    Freeze all parameters $\mathbb{O}$
    Pick next "suboptimal" leaf node $l \in \mathcal{N}_{leaf}$ in the breadth-first order
    Add (1) router to $l$ and train new parameters          $\triangleright$ Split data
    Add (2) transformer to the incoming edge of $l$ and train new parameters          $\triangleright$ Deepen transform
    Add (1) or (2) permanently to $\mathbb{T}$ if validation error decreases, otherwise leaf is set to "optimal"
    Add any new modules to $\mathbb{O}$
    **if** no "suboptimal" leaves remain **then**
        Break
Unfreeze and train all parameters in $\mathbb{O}$          $\triangleright$ Global refinement with fixed $\mathbb{T}$

---

## B  ADDITIONAL RELATED WORK

The tree-structure of ANTs naturally performs conditional computation. We can also view the proposed tree-building algorithm as a form of neural architecture search. Here we provide surveys of these areas and their relations to ANTs.

**Conditional Computation:** In NNs, computation of each sample engages every parameter of the model. In contrast, DTs route each sample to a single path, only activating a small fraction of the model. Bengio Bengio (2013) advocated for this notion of conditional computation to be integrated into NNs, and this has become a topic of growing interest. Rationales for using conditional computation ranges from attaining better capacity-to-computation ratio (Bengio et al., 2013; Davis & Arel, 2013; Bengio et al., 2015; Shazeer et al., 2017) to adapting the required computation to the difficulty of the input and task (Bengio et al., 2015; Almahairi et al., 2016; Teerapittayanon et al., 2016; Graves, 2016; Figurnov et al., 2017; Veit & Belongie, 2017). We view the growth procedure of ANTs as having a similar motivation with the latter—processing raw pixels is suboptimal for computer vision tasks, but we have no reason to believe that the hundreds of convolutional layers in current state-of-the-art architectures (He et al., 2016; Huang et al., 2017) are necessary either. Growing ANTs adapts the architecture complexity to the dataset as a whole, with routers determining the computation needed on a per-sample basis.

**Neural Architecture Search:** The ANT growing procedure is related to the progressive growing of NNs (Fahlman & Lebiere, 1990; Hinton et al., 2006; Xiao et al., 2014; Chen et al., 2016; Srivastava et al., 2015; Lee et al., 2017; Cai et al., 2018; İrsoy & Alpaydın, 2018), or more broadly, the field of neural architecture search (Zoph & Le, 2017; Brock et al., 2017; Cortes et al., 2017). This approach, mainly via greedy layerwise training, has historically been one solution to optimising NNs (Fahlman & Lebiere, 1990; Hinton et al., 2006). However, nowadays it is possible to train NNs in an end-to-end fashion. One area which still uses progressive growing is lifelong learning, in which a model needs to adapt to new tasks while retaining performance on previous ones (Xiao et al., 2014; Lee et al., 2017). In particular, (Xiao et al., 2014) introduced a method that grows a tree-shaped network to accommodate new classes. However, their method never transforms the data before passing it to the children classifiers, and hence never benefit from the parent's representations.

Whilst we learn the architecture of an ANT in a greedy, layerwise fashion, several other methods search globally. Based on a variety of techniques, including evolutionary algorithms (Stanley & Miikkulainen, 2002; Real et al., 2017), reinforcement learning (Zoph & Le, 2017), sequential optimisation (Liu et al., 2017) and boosting (Cortes et al., 2017), these methods find extremely high-performance yet complex architectures. In our case, we constrain the search space to simple tree-structured NNs, retaining desirable properties of DTs such as data-dependent computation and interpretable structures, while keeping the space and time requirement of architecture search tractable thanks to the locality of our growth procedure.

## C  TRAINING DETAILS

We perform our experiments on the MNIST digit classification task (LeCun et al., 1998) and CIFAR-10 object recognition task (Krizhevsky & Hinton, 2009). The MNIST dataset consists of $60,000$ training and $10,000$ testing examples, all of which are $28 \times 28$ grayscale images of digits from $0$ to $9$ (10 classes). The dataset is preprocessed by subtracting the mean, but no data augmentation is used. The CIFAR-10 dataset consists of $50,000$ training and $10,000$ testing examples, all of which are $32 \times 32$ coloured natural images drawn from 10 classes. We adopt an augmentation scheme widely used in the literature (Goodfellow et al., 2013; Lin et al., 2014; Springenberg et al., 2015; He et al., 2016; Huang et al., 2017) where images are zero-padded with 4 pixels on each side, randomly cropped and horizontally mirrored.

For both datasets, we hold out $10\%$ of training images as a validation set. The best model is selected based on the validation accuracy over the course of ANT training, spanning both the growth phase and the refinement phase, and its accuracy on the testing set is reported. The hyperparameters are also selected based on the validation performance alone.

Both the growth and refinement phase of ANTs takes up to 2 hours on a single Titan X GPU on both datasets. For all the experiments in this paper, we employ the following training protocol: (1) optimize parameters using Adam (Kingma & Ba, 2014) with initial learning rate of $10^{-3}$ and $\beta = [0.9, 0.999]$, with minibatches of size $512$; (2) during the growth phase, employ early stopping with a patience of 5, that is, training is stopped after 5 epochs of no progress on the validation set; (3) during the refinement phase, train for $100$ epochs for MNIST and $200$ epochs for CIFAR-10, decreasing the learning rate by a factor of 10 at every multiple of 50.

## D  TRAINING TIMES

Tab. 5 summarises the time taken on a single Titan X GPU for the growth phase and refinement phase of various ANTs, and compares against the training time of All-CNN (Springenberg et al., 2015). Local optimisation during the growth phase means that the gradient computation is constrained to the newly added component of the graph, allowing us to grow a good candidate model under 2 hours on a single GPU.

Table 5: Training time comparison. Time and number of epochs taken for the growth and refinement phase are shown. along with the time required to train the baseline, All-CNN (Springenberg et al., 2015).

| Model | Growth | | Fine-tune | |
|---|---|---|---|---|
| | Time | Epochs | Time | Epochs |
| All-CNN (baseline) | – | – | 1.1 (hr) | 200 |
| ANT-CIFAR10-A | 1.3 (hr) | 236 | 1.5 (hr) | 200 |
| ANT-CIFAR10-B | 0.8 (hr) | 313 | 0.9 (hr) | 200 |
| ANT-CIFAR10-C | 0.7 (hr) | 285 | 0.8 (hr) | 200 |

## E  EFFECT OF TRAINING STEPS IN THE GROWTH PHASE

Fig. 4 compares the validation accuracies of the same ANT-CIFAR-C model trained on the CIFAR-10 dataset with varying levels of patience during early stopping in the growth phase. A higher patience level corresponds to more training epochs for optimising new modules in the growth phase. When the patience level is 1, the architecture growth terminates prematurely and plateaus at low accuracy at $80\%$. On the other hand, a patience level of 15 causes the model to overfit locally with $87\%$. In between these, the patience level of 5 gives the best results with $91\%$ validation accuracy.

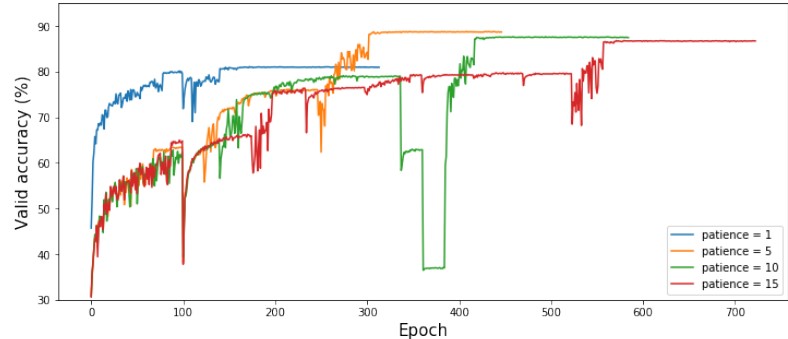

Figure 4: Effect of patience level on the validation accuracy trajectory during training. Each curve shows the validation accuracy on CIFAR-10 dataset.

## F  EXPERT SPECIALISATION

We investigate if the learned routing strategy is meaningful by comparing the classification accuracy of our default path-wise inference against that of the predictions from the leaf node with the smallest reaching probability. Tab. 6 shows that using the least likely "expert" leads to a substantial drop in classification accuracy, down to close to that of random guess or even worse for large trees (ANT-MNIST-C and ANT-CIFAR10-C). This demonstrates that ANTs have the capability to split the input space in a meaningful way.

Table 6: Comparison of classification performance between the default single-path inference scheme and the prediction based on the least likely expert. between the

| Module Spec. | Error % (Selected path) | Error % (Least likely path) |
|---|---|---|
| ANT-MNIST-A | 0.69 | 86.18 |
| ANT-MNIST-B | 0.73 | 81.98 |
| ANT-MNIST-C | 1.68 | 98.84 |
| ANT-CIFAR10-A | 8.32 | 74.28 |
| ANT-CIFAR10-B | 9.18 | 89.74 |
| ANT-CIFAR10-C | 9.34 | 97.52 |

## G  VISUALISATION OF DISCOVERED ARCHITECTURES

Fig. 5 shows ANT architectures discovered on the MNIST (i-iii) and CIFAR-10 (iv-vi) datasets. We observe two notable trends. Firstly, most architectures learn a few levels of features before resorting to primarily splits. However, over half of the architectures (ii-v) still learn further representations beyond the first split. Secondly, all architectures are unbalanced. This reflects the fact that some groups of samples may be easier to classify than others. This property is reflected by traditional DT algorithms, but not "neural" tree-structured models that stick to pre-specified architectures (Laptev & Buhmann, 2014; Frosst & Hinton, 2017; Kontschieder et al., 2015; Ioannou et al., 2016).

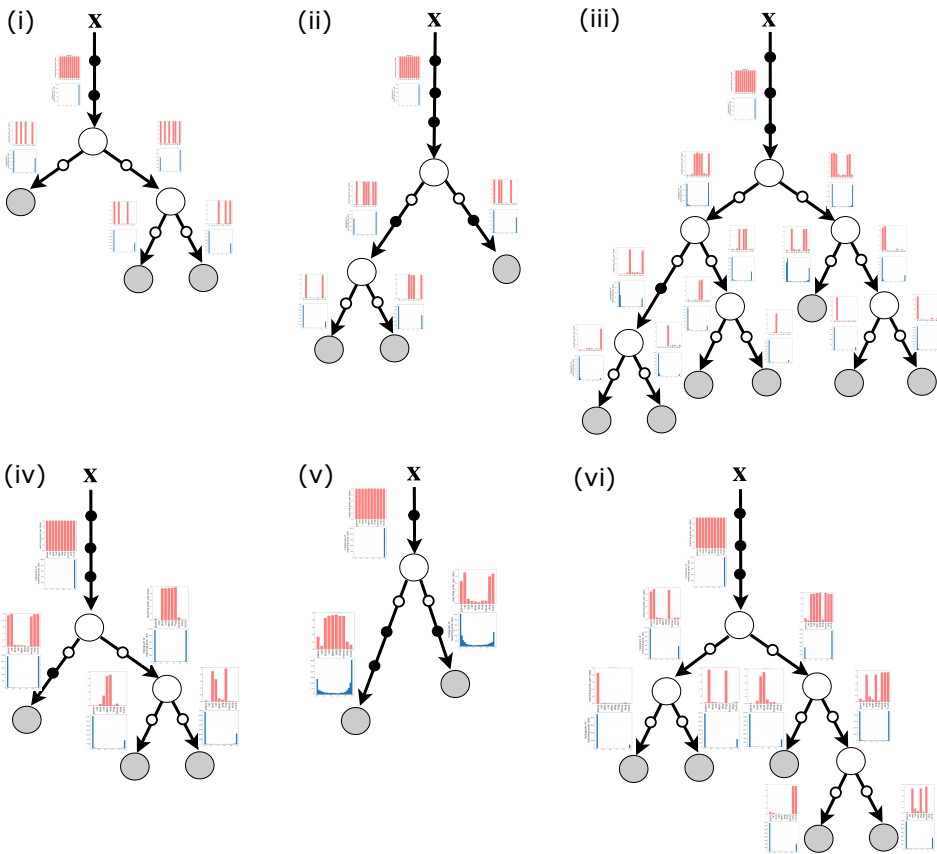

Figure 5: Illustration of discovered ANT architectures. (i) ANT-MNIST-A, (ii) ANT-MNIST-B, (iii) ANT-MNIST-C, (iv) ANT-CIFAR10-A, (v) ANT-CIFAR10-B, (vi) ANT-CIFAR10-C. Histograms in red and blue show the class distributions and path probabilities at respective nodes. Small black circles on the edges represent transformers, circles in white at the internal nodes represent routers, and circles in gray are solvers. The small white circles on the edges denote specific cases where transformers are identity functions.

## H MULTIVARIATE REGRESSION

The ANT algorithm is general purpose, and can be applied to problems other than classification on image data. To demonstrate this, we also grow ANTs to perform (multivariate) regression on the SARCOS robot inverse dynamics dataset[2], which consists of 44,484 training and 4,449 testing examples, where the goal is to map from the 21-dimensional input space (7 joint positions, 7 joint velocities and 7 joint accelerations) to the corresponding 7 joint torques (Vijayakumar & Schaal, 2000). No dataset preprocessing or augmentation is used. We hold out 10% of the training examples as a validation set. Baseline MLPs, routers and transformers are composed of single fully connected layers with 256 units with tanh nonlinearities, and the solver is a linear regressor. Other training details are the same as for classification (see Supp. Sec. C). All non-NN-based methods were trained using scikit-learn (Pedregosa et al., 2011); only single-output GBT models were available so 7 separate GBTs were trained.

The results are shown in Tab. 7. ANT-SARCOS outperforms all other methods in mean squared error with the full set of parameters, with GBTs performing slightly better using single-path inference. In comparison with results on MNIST and CIFAR-10, we note that the top 3 performing methods are all tree-based, with the third best method being an SDT (with MLP routers). This highlights the power of splitting the input space and conditional computation, both of which standard NNs are not capable of. Meanwhile, we still reap the benefits of representation learning, as shown by both ANT-SARCOS and the SDT (which is a specific form of ANT) requiring fewer parameters than the best-performing GBT configuration. Finally, we note that deeper NNs (5 vs. 3 hidden layers) can overfit on this small dataset, which makes the adaptive growth procedure of tree-based methods ideal for finding a model that exhibits good generalisation.

Table 7: Comparison of performance of different models on SARCOS. The columns "Error (Full)" and "Error (Path)" indicate the mean squared error of predictions based on the full distribution and the single-path inference. The columns "Params. (Full)" and "Params. (Path)" respectively show the total number of parameters in the model and the average number of parameters utilised during single-path inference. "Ensemble Size" indicates the size of ensemble used to attain the reported accuracy. Results from Zhao et al. (2017) are included as a reference value from prior work, but are not directly comparable as they hold out 30% of the training examples as a validation set.

| | Method | Error (Full) | Error (Path) | Params. (Full) | Params. (Path) | Ensemble Size |
|---|---|---|---|---|---|---|
| SARCOS | Linear regression | 10.693 | N/A | **154** | N/A | 1 |
| | MLP with 2 hidden layers (Zhao et al., 2017) | 5.111 | N/A | 31,804 | N/A | 1 |
| | Decision tree | 3.708 | 3.708 | 319,591 | **25** | 1 |
| | MLP with 1 hidden layer | 2.835 | N/A | 7,431 | N/A | 1 |
| | Gradient boosted trees | 2.661 | 2.661 | 391,324 | 2,083 | 7 × 30 |
| | MLP with 5 hidden layers | 2.657 | N/A | 270,599 | N/A | 1 |
| | Random forest | 2.426 | 2.426 | 40,436,840 | 4,791 | 200 |
| | Random forest | 2.394 | 2.394 | 141,540,436 | 16,771 | 700 |
| | MLP with 3 hidden layers | 2.129 | N/A | 139,015 | N/A | 1 |
| | SDT (with MLP routers) | 2.118 | 2.246 | 28,045 | 10,167 | 1 |
| | Gradient boosted trees | 1.444 | **1.444** | 988,256 | 6,808 | 7 × 100 |
| | ANT-SARCOS | **1.384** | 1.542 | 103,823 | 61,640 | 1 |

**Ablation study:** we compare the regression error of our ANT in cases where the options for adding transformer or router modules are disabled (see Tab. 8). In this experiment, patience levels are tuned separately for respective models. In the first case, the resulting models are equivalent to SDTs (Suárez & Lutsko, 1999) or HMEs (Jordan & Jacobs, 1994) with locally grown architectures, while the second case is equivalent to standard NNs, grown adaptively layer by layer. We observe that either ablation consistently leads to higher regression errors across different module configurations.

Table 8: Ablation study to compare the effects of different components of ANTs on regression performance. "NN" refers to the case where the ANT is grown without routers while "SDT/HME" refers to the case where transformer modules on the edges are disabled.

| Module Spec. | Error (Full) | | | Error (Path) | | |
|---|---|---|---|---|---|---|
| | ANT (default) | NN (no routers) | SDT/HME (no transformers) | ANT (default) | NN (no routers) | SDT/HME (no transformers) |
| ANT-SARCOS | 1.384 | 2.511 | 2.118 | 1.542 | 2.511 | 2.246 |

---

[2]http://www.gaussianprocess.org/gpml/data/

# I ENSEMBLING

As with traditional DTs (Breiman, 2001) and NNs (Hansen & Salamon, 1990), ANTs can be ensembled to gain improved performance. In Tab. 9 we show the results of ensembling 8 ANTs (using the "-A" configurations for classification), each of which is trained with a randomly chosen split between training and validation sets. We compare against the single tree models, trained with the default split as used in the training of models reported in Tab. 3 and Tab. 7. In all cases both the full and single-path inference performance is noticeably improved, and in MNIST we reach close to state-of-the-art performance (0.29% versus 0.25% (Sabour et al., 2017)) with significantly fewer parameters (851k versus 8.2M; see Tab. 10 for ensemble and Tab. 3 for baseline parameter counts).

Table 9: Comparison of prediction errors of a single ANT versus an ensemble of 8, with predictions averaged over all ANTs in the ensemble.

|  | MNIST (Class Error %) | | CIFAR-10 (Class Error %) | | SARCOS (MSE) | |
|---|---|---|---|---|---|---|
|  | Error (Full) | Error (Path) | Error (Full) | Error (Path) | Error (Full) | Error (Path) |
| Single model | 0.64 | 0.69 | 8.31 | 8.32 | 1.384 | 1.542 |
| Ensemble | 0.29 | 0.30 | 7.76 | 7.79 | 1.226 | 1.372 |

Table 10: Parameter counts for a single ANT versus an ensemble of 8.

|  | MNIST | | CIFAR-10 | | SARCOS | |
|---|---|---|---|---|---|---|
|  | Params. (Full) | Params. (Path) | Params. (Full) | Params. (Path) | Params. (Full) | Params. (Path) |
| Single model | 100,596 | 84,935 | 1.4M | 1.0M | 103,823 | 61,640 |
| Ensemble | 850,775 | 655,449 | 8.7M | 7.4M | 598,280 | 360,766 |

