# OpenReview forum: "Adaptive Neural Trees"
_ICLR.cc/2019/Conference_

### Official Review · AnonReviewer2 · 2018-10-29
**acceptable**

**Rating:** 6
**Confidence:** 3

**Review:**


The authors proposed a new model Adaptive Neural Trees(ANTs) by combining the representation learning and gradient optimization of neural networks with architecture learning of decision trees. The key advantage of the new model ANTs  over the existing methods(Random forest, Linear classifier, Neural decision forest, et al) is: it may achieve high accuracy(above $90\%$) with relatively much smaller number of parameters, as shown by the experiments on the datasets MNIST and CIFAR-10. Besides, the authors proposed single-path inference based on the greedily-selected leaf node to approximate the multi-path inferences with the full predictive distribution. The experiments show the single-path inference doesn't lose much accuracy but it saves memory and time. This paper is acceptable after minor modification.


Questions:
In the second line below equation (1), $n$ in $t_{e_{n(j)}}^{\psi}$ is not defined. Also, should $t_{e_{1}}^{\psi}$ be $t_{e_{n(1)}}^{\psi}$?

---

> ### Author Response · Authors · 2018-11-08
> **Response to Reviewer 2**
>
> Thank you very much for your review and proposed corrections. We have updated the paper with the added definition and corrected the equation accordingly.

---

> ### Author Response · Authors · 2018-12-14
> **Additional experiments**
>
>
> We would like to let you know that we have updated our work by including the following results:
>
> 1. A full set of results on the SARCOS robot inverse dynamics dataset for multivariate regression in Supp. Sec. H. (page 19) to show the wider applicability of ANTs.
>
> 2. Results from ensembling ANTs (Supp. Sec. I, page 20), demonstrating improved performance on all 3 datasets, including close to state-of-the-art performance on MNIST with an order of magnitude fewer parameters.
>
> We have noticed that the overall score has been dropped from 7 to 6. We would be grateful if you could let us know any concerns you might have that we could address.

---

### Official Review · AnonReviewer3 · 2018-11-05
**limited experiments, doubts about the method**

**Rating:** 6
**Confidence:** 4

**Review:**

The paper is written rather well, however I find the experiments incomplete and have some reservations about the
method. My main points of critique are:

1.  Combining DT & NN
I have doubts that the way you combine DT &NN  you get the "Best of both world". In some ways your architecture also
shares disadvantages of both:

1.1 Interpretability
Because each node in the tree can a neural network (with arbritrary complexity), this approach looses one central advantage of DT, that is the interpretability of the result.    Each node in the tree can perform arbritrary complex (and hierarchical)
computations. The authors only show one particular example (Fig. 2a), where the model has learned is a reasonable
structure.

1.2 Complexity:
The whole architecture is much more complex than either a neural network or a decision tree. I expect that therefore training these is not easy, and expert knowledge in either DT  or NN may not be enough to use this model.


2. Limited experiments

2.1 The authors only consider 2 experiments from vision (MNIST & CIFAR 10) while proposing a universal method.  To show universality the authors should use data sets from different domains (e.g UCI data sets)

2.2 The authors argue that a  strength of the method   is  that it uses a low number of parameters on average for a forward path (compared to the total parameter size).  I don't find this argument to be convincing. In the limit this would imply a high memorization of the  data.  Also, a similar case can be made for standard CNN, when a particular filter is mostly inactive for some data points.

2.3 The interpretability of DT compared to NN I mentioned earlier.  To make the argument that their method learns the
hierarchical structure of the data , the authors should have added experiments to support this, where  such a hierarchical structure is clearly present and can be evaluated empirically.


--

In light of the extended experiments w.r.t. to 2.1 I increased my score from 5 to 6.  Overall, I still have doubts about the interpretability and complexity of the proposed method.

Complexity:  "but all the intuitions needed would come solely from training NN".    I disagree with this response.   The architecture is a mix between a tree (hard, decision-tree like error surface,  non-local) and neural network (smooth, mostly convex error surface). This also implies that the training process and its behavior will possess patterns and challenges of both approaches.

Interpretability:  I think the method misses "priors" that enforce credit assignment.  Partitioning the problem in subp-roblems should be done via the tree components, whereas processing (such as image filtering) should be done in the network nodes. However,  the method does not enforce, or encourage this behavior, for instance
via constraints:   also nodes can do partitioning (because neural networks can approximate decision trees)  and edges can do processing (e.g. decisions-trees can be used for mnist).

So I still believe this to be a borderline paper, however, the experiments support a more general applicability.

---

> ### Author Response · Authors · 2018-11-09
> **Response to Reviewer 3**
>
>
> Thank you for your critical feedback. We hope to interact with you here to fully clarify and address your concerns and questions. Below are our initial responses to your comments.
>
> 1.1 & 2.3 (interpretability)
> ANTs add representational power in exchange for some interpretability (but ANTs do not lose any representational power compared to standard NNs). However, we believe that the hierarchical structure of the model still provides a new means for interpreting its decision making process. As noted in the paper, we can observe exemplar-based hierarchies in which every routing decision divides certain classes/categories of data; this relates to other interpretability methods such as finding images which maximally activate a given neuron [1]. This could, for example, be used to localise a point of routing failure if an ANT makes the wrong decision during inference. As you note, a similar specialisation can occur in certain filters in conventional CNNs, but the fully distributed representation makes localisation of failures very difficult.
>
> Regarding the evaluation of the quality of hierarchical structures, we are unaware of any quantitative metric that can be used when the ground truth is unknown or ambiguous; for instance, we know that there is no hierarchical generative process behind the construction of the MNIST digits dataset. More importantly, we think that human intuitions on relevant hierarchical structures do not necessarily equate to optimal clustering of data and thus such quantification might be quite difficult. Rather, what we would like to highlight is the ability of ANTs to learn when to share or separate the representation of data to optimise end-task performance, which gives rise to automatically discovering such hierarchies (such as the cherry-picked example in Fig. 2).
>
> 1.2 (complexity)
> The complexity of tuning ANTs is indeed more than that of NNs or DTs, but all the intuitions needed would come solely from training NNs.
>
> 2.2. (memorization)
> We are unsure if we have understood your criticism, so please do let us know if we do not correctly address your concerns. A common definition of memorisation is that of a model overfitting to the training set and hence failing to generalise to the validation and test sets [2]. Because the tree growth mechanism utilises early stopping [3], ANTs adapt their complexity to the training dataset and generalise well, even when trained on very small datasets (Fig 3. left).
>
> We perceive the high performance of greedy routing as the appropriate specialisation/construction of a hierarchy of experts [4], where the final branches/leaves only need to account for specific partitions of the data; see also Table 6 for an ablation where inverting routing reduces performance catastrophically. Even if this specialisation can occur in certain filters in a conventional CNN, conventional CNNs lack the explicit hierarchy of ANTs. Another benefit of conditional computation, as used in ANTs, is that single-path inference is lightweight [5], as opposed to traditional NNs, which require engaging every parameter during the forward pass.
>
> [1] Girshick, R., Donahue, J., Darrell, T., & Malik, J. (2014). Rich feature hierarchies for accurate object detection and semantic segmentation. In Proceedings of the IEEE conference on computer vision and pattern recognition (pp. 580-587).
> [2] Arpit, D., Jastrzębski, S., Ballas, N., Krueger, D., Bengio, E., Kanwal, M. S., ... & Lacoste-Julien, S. (2017). A closer look at memorization in deep networks. Proceedings of Machine Learning Research, 70.
> [3] Prechelt, L. (1998). Early stopping-but when?. In Neural Networks: Tricks of the trade (pp. 55-69). Springer, Berlin, Heidelberg.
> [4] Jordan, M. I., & Jacobs, R. A. (1994). Hierarchical mixtures of experts and the EM algorithm. Neural computation, 6(2), 181-214.
> [5] Bengio, E., Bacon, P. L., Pineau, J., & Precup, D. (2015). Conditional computation in neural networks for faster models. arXiv preprint arXiv:1511.06297.

---

> ### Author Response · Authors · 2018-11-26
> **Additional Experiments to Address 2.1**
>
> 2.1 (universality)
> We thank you for suggesting that we provide empirical evidence on different sorts of data/domains to demonstrate the universality of our method. To this end we have now included a full set of results on the SARCOS robot inverse dynamics dataset for multivariate regression [1] in Supp. Sec. H. (page 19). Using the same training setup we have produced results for ANTs as well as a large variety of baselines. This is a challenging dataset for standard NNs, with the 3 top-performing methods all being tree-based: SDTs (a specific form of ANTs), GBTs and ANTs. The best configuration for GBTs requires an order more parameters than the ANT, so taken together, these results (see Tab. 7) show the benefits of both representation learning and partitioning of the data space. We also perform the same ablation study that we did for MNIST and CIFAR-10, and show that, again, both nonlinear routers and transformers are key components of the ANT algorithm (see Tab. 8).
>
> Additionally, we have also included results from ensembling ANTs (Supp. Sec. I, page 20), demonstrating improved performance on all 3 datasets, including close to state-of-the-art performance on MNIST [2] with an order of magnitude less parameters.
>
> [1] Vijayakumar, S., & Schaal, S. (2000, June). Locally weighted projection regression: An o (n) algorithm for incremental real time learning in high dimensional space. In Proceedings of the Seventeenth International Conference on Machine Learning (ICML 2000) (Vol. 1, pp. 288-293).
> [2] Sabour, S., Frosst, N., & Hinton, G. E. (2017). Dynamic routing between capsules. In Advances in Neural Information Processing Systems (pp. 3856-3866).

---

> ### Author Response · Authors · 2018-12-14
> **Further responses**
>
>
> Thank you for raising your score in recognition of our efforts with extending our task set. We have some further responses:
>
> Complexity:
> We agree, and apologise for our misinterpretation.  From a pragmatic perspective, we just wanted to explain that many decisions made in the implementation of ANTs in our case (e.g. choices of primitive modules, optimiser and its scheduling, etc) were based on accepted practices in deep learning research. From a theoretical perspective, we agree that as a natural consequence of attempting to combine the two approaches, improving the optimisation of ANTs would face challenges of both, and we believe this would be a worthwhile research question to pursue.
>
> Interpretability:
> The tree-topology is a strong structural prior that enforces a particular sparse structure by which features are shared and separated in a hierarchical fashion. Such sparsity might approximately arise in a feedforward neural networks, but is not explicitly enforced and is difficult to track. For example, in our work, the tree structure approximately partitioned the input space into disjoint subsets (mostly grouping together images of certain classes), each of which is processed by a separate neural network whose early layers are more shared and deeper layers are more specific to respective subsets.
>
> However, that said, our proposed method learns such sparsity based on validation loss. As you point out, I believe if you have good “prior” knowledge about what might constitute “good” splits (e.g. purity of classes, known hierarchical structures in data, etc), encouraging the tree-structure to account for it would benefit the performance/interpretability of the resultant model.

---

### Official Review · AnonReviewer1 · 2018-11-05
**This paper proposes the Adaptive Neural Trees (ANT) approach to combine the two learning paradigms of deep neural nets and decision trees (DT).**

**Rating:** 4
**Confidence:** 4

**Review:**

The presented method is a generalization of a number of existing methods, which can be regarded as special cases. Overall the method seems to be novel. Meanwhile, I have two major questions:
To account for the bias issue, instead of a single DT, ensemble methods such as random forests are the popular choices. How ANT could benefit from relying on a single DT instead of a random forest type?
The datasets of MNIST and CIFAR-10 are used for many years and the performance is already saturated. As presented in Table.3, the performance of the proposed method is also not the best on either of the tested datasets. Please clearly elaborate on why and how to address this issue. It would be more interesting and meaningful to work with a more recent large datasets, such as ImageNet or MS COCO.

The response does not fully address my concerns.

---

> ### Author Response · Authors · 2018-11-08
> **Response to Reviewer 1**
>
> Thank you for your review. We would be grateful if we could engage in a constructive discussion to address your questions.
>
> For your first point, due to the representation learning properties of ANTs, they are more similar to NNs than DTs in that they are more prone to overfitting than bias. We mitigate overfitting by utilising early stopping on validation error. We have also shown that the progressive growth of the architecture defends against overfitting on small training datasets (Fig. 3). That said, ensembles of neural networks (NNs) have also long been used for reducing bias [1], so we could equally train an ensemble of ANTs to mimic the random forest approach.
>
> For your second point, we would like to note that different models have different trade-offs - Gaussian processes are extremely data efficient, but scale poorly to large datasets, while NNs have the opposite properties. Our goal with ANTs is not to compete with the the large amount of research into optimising architectures and regularisation methods that are needed to get state-of-the-art results, but show that our proposed model is able to achieve reasonably high accuracies whilst retaining the important properties of DTs; (1) ability to learn hierarchical clustering of data, and (2) the data-driven architectures. This is in line with other research into novel architectures, such as capsule networks [2], or NN-hybrids such as neural processes [3], which have been evaluated on smaller datasets.
>
> [1] Hansen, L. K., & Salamon, P. (1990). Neural network ensembles. IEEE transactions on pattern analysis and machine intelligence, 12(10), 993-1001.
> [2] Sabour, S., Frosst, N., & Hinton, G. E. (2017). Dynamic routing between capsules. In Advances in Neural Information Processing Systems (pp. 3856-3866).
> [3] Garnelo, M., Schwarz, J., Rosenbaum, D., Viola, F., Rezende, D. J., Eslami, S. M., & Teh, Y. W. (2018). Neural processes. In International Conference on Machine Learning.

---

> ### Author Response · Authors · 2018-11-26
> **Additional Experiments**
>
>
> In addition to our earlier response, to empirically address your first point, we have now included results from ensembling ANTs (Supp. Sec. I, page 20), demonstrating improved performance on all 3 datasets, including close to state-of-the-art performance on MNIST [1] with an order of magnitude fewer parameters.
>
> We have also now included a full set of results on the SARCOS robot inverse dynamics dataset for multivariate regression [2] in Supp. Sec. H. (page 19) to demonstrate that our method can work on different domains/tasks. Using the same training setup we have produced results for ANTs as well as a large variety of baselines. This is a challenging dataset for standard NNs, with the 3 top-performing methods all being tree-based: SDTs (a specific form of ANTs), GBTs and ANTs. The best configuration for GBTs requires an order more parameters than the ANT, so taken together, these results (see Tab. 7) show the benefits of both representation learning and partitioning of the data space. We also perform the same ablation study that we did for MNIST and CIFAR-10, and show that, again, both nonlinear routers and transformers are key components of the ANT algorithm (see Tab. 8).
>
> As the reviewer points out, experiments on larger datasets (e.g. ImageNet) would be more interesting and useful for showing scalability.  However, we believe that our results on 2 image datasets (MNIST, CIFAR-10) and 1 non-image dataset (SARCOS) on both classification and regression tasks are sufficiently compelling to motivate such future work.
>
> [1] Sabour, S., Frosst, N., & Hinton, G. E. (2017). Dynamic routing between capsules. In Advances in Neural Information Processing Systems (pp. 3856-3866).
> [2] Vijayakumar, S., & Schaal, S. (2000, June). Locally weighted projection regression: An o (n) algorithm for incremental real time learning in high dimensional space. In Proceedings of the Seventeenth International Conference on Machine Learning (ICML 2000) (Vol. 1, pp. 288-293).

---

### Public Comment · ~Buu_Phan1 · 2018-11-10
**Nice paper**

I think this is a very cool paper. I just wonder if you have an open-source code for this?

---

> ### Author Response · Authors · 2018-11-13
> **Code release**
>
> Thank you for your interest in our paper. We will release our code upon acceptance and link to it within the paper.

---

### Meta-Review · Area_Chair1 · 2018-12-10
**metareview: unconvincing experiments**

**Confidence:** 5
**Recommendation:** Reject

**Metareview:**

This paper proposes adaptive neural trees (ANT), a combination of deep networks and decision tress. Reviewers 1 leans toward reject the paper, pointing out several flaws. Reviewer 3 also raises concerns, despite later increasing rating to marginally above threshold.  Of particular note is the weak experimental validation.

The paper reports results only on MNIST and CIFAR-10. MNIST performance is too easily saturated to be meaningful. The CIFAR-10 results show ANT models to have far greater error than the state-of-the-art deep neural network models.

As Reviewer 1 states, "performance of the proposed method is also not the best on either of the tested datasets. Please clearly elaborate on why and how to address this issue. It would be more interesting and meaningful to work with a more recent large datasets, such as ImageNet or MS COCO."

The rebuttal fails to offer the type of additional results that would remedy this situation. Without a convincing experimental story, it is not possible to recommend acceptance of this paper.